# Nutrition in early life interacts with genetic risk to influence preadult behaviour in the Raine Study

Lars Meinertz Byg [1,2], Carol A. Wang [1,2], John Attia [3], Andrew J. O. Whitehouse[4], Wendy H. Oddy[5], Jonathan J. Hirst [1] & Craig E. Pennell [1,2,6] ✉

## Abstract

**Background:** Early life nutrition is associated with child behaviour; however, the interplay with genetic vulnerability is understudied. We hypothesised that psychiatric genetic risk interacted with early nutrition to predict behavioural problems in childhood and adolescence. **Methods:** The Raine Study participants with genetic information aged 2–17 were repeatedly evaluated with the child behaviour checklist total problems score (CBCL$_{TOT}$). Breastfeeding duration was recalled at age 1, 2 and 3 follow-up, and toddler diet derived by an age-1 24-h maternal recall (EAT1, scale 0–70, SD 10, higher scores proxying healthy diet). We derived polygenic scores (PGS) impacting general psychopathology: attention-deficit hyperactivity disorder (ADHD), depression, chronic multisite pain (CMSP), total behaviour problems and birthweight. In confounder-adjusted mixed-effects models of CBCL$_{TOT}$ throughout follow-up we examined nutrition-by-PGS interactions. **Results:** In 1393 participants, a borderline signal suggests that 1 month longer breastfeeding reduces CBCL$_{TOT}$ by −0.108 (95% CI [−0.184, −0.0289]) exclusively in individuals with a higher CMSP PGS (Interaction $p = 0.03$). In 1310 participants, a strong signal suggests that 1 EAT1 point increase results in a reduced CBCL$_{TOT}$ by 0.121 points (95% CI [−0.171, −0.0704]) exclusively in individuals with a lower ADHD PGS (Interaction $p = 0.0005$). *Post hoc* analysis suggests that plant-based food consumption drives the favourable EAT1-CBCL$_{TOT}$ association. **Conclusions:** Nutrition in early life and psychiatric genetic risk may interact to determine lasting child behaviour. Contrary to our hypothesis, we find dietary benefits in individuals with lower ADHD PGS, necessitating replication. We also highlight the possibility of including genetics in early nutrition intervention trials for causal inference.

## Plain language summary

It has been proposed that both early nutrition and genetic risk influence behaviour in childhood and adolescence. However, the interplay between the two remains poorly understood. We examined the interaction between genetic and early nutritional data with repeated parent ratings of behaviour from birth until age 17 years in Australian children. We saw fewer behavioural problems in those with a healthy diet at one year of age if the children had a low risk of developing ADHD based on their genetics. In contrast, longer breastfeeding duration primarily reduced behavioural problems in those with a higher risk of chronic pain based on their genetics. Our results suggest that tailoring early nutrition based on genetic risk could reduce pre-adult behaviour problems. However this association should be further tested in clinical studies.

Psychiatric illnesses are among the top contributors to disability globally[1] and arise from an interaction between genes and the environment (GxE)[2,3] at critical—often early-periods of development (GxExTime).[4] Mental illness can manifest in childhood with behavioural problems[5,6] and childhood behaviour is similarly subject to GxE interactions.[7] The stratification of individuals based on genetics could potentially identify those eligible for early intervention in the form of environmental modifications.[8] In a clinical context the ever-decreasing cost of high-throughput analyses and the potential availability of genetic material from routine heel-prick[9] implies such precision prevention could be within reach;[8] however, potential genotypes and beneficial interventions are still subject to investigation.

One modifiable environmental exposure consistently associated with child behaviour is nutrition[10,11] and evidence suggests that it is also possible to change early diet through individual[12,13] and community intervention.[14]

[1]Mothers and Babies Research Center, Hunter Medical Research Institute, Newcastle, NSW, Australia. [2]School of Medicine and Public Health, University of Newcastle, Newcastle, NSW, Australia. [3]Department of General Medicine, John Hunter Hospital, Newcastle, NSW, Australia. [4]The Kids Research Institute Australia, University of Western Australia, Perth, WA, Australia. [5]Menzies Institute for Medical Research, University of Tasmania, Hobart, TAS, Australia. [6]Department of Maternity and Gynaecology, John Hunter Hospital, Newcastle, NSW, Australia. ✉e-mail: Craig.Pennell@newcastle.edu.au

In the Raine Study, both breastfeeding[15] and Mediterranean diet[16] have been associated with reduced problem behaviour in childhood and adolescence; however, traditional observational studies do not account for potential genetic confounding of perinatal risks (including poor nutrition) or GxE-dependent effects.[17] For instance there is poor consistency between observational studies and results from the large breastfeeding trial PROBIT.[18,19] Here breastfeeding support programs increased cognition at age 6.5 but did not reduce problem behaviour,[12] which could suggest specific confounding in nutrition-behaviour correlates. Alternatively early nutritional effects on long-term outcomes may vary by underlying genotype as demonstrated for cardiometabolic health,[20] meaning interventions should be directed to at-risk populations.

Studies of gene-nutrition interactions in determining child behaviour have been limited by sparse results in the genome-wide association studies (GWAS) of child behaviour, albeit the most recent efforts yielded two significant genetic variants associated with total problems;[21] however the co-morbidity patterns and changing manifestations of psychopathology across childhood and adolescence have prompted research into cross-disorder prediction from psychiatric GWAS.[22] A recent screening of genetic risk profiles demonstrated that polygenic scores (PGS) for attention-deficit hyperactivity disorder (ADHD),[23] depression[24] and chronic multisite pain (CMSP)[25] had considerable influence on a construct of general child psychopathology[26] derived from the parent-reported child behaviour checklist (CBCL).[27] The utility of these four genetic risks in predicting early nutrition effects on behaviour has not been examined. Surprisingly, an association has also emerged between birth weight (BW) PGS and childhood behaviour[28,29] with different effects in males and females,[30] and our group previously demonstrated an interaction between BW-PGS and early nutrition in predicting adult cardiometabolic risk.[20] The two exposures of interest (breastfeeding and early diet) were therefore screened for interaction with five PGS to predict behaviour problems in childhood and adolescence. We hypothesised that breastfeeding and diet effects on problem behaviour would be larger in genetically vulnerable individuals.

We show that healthier year one diet is associated with reduced $CBCL_{TOT}$ in those with a lower PGS of ADHD driven primarily by plant-based food intake, with reduced diet effect size at age three; in contrast, a longer breastfeeding duration was associated with reduced $CBCL_{TOT}$ in those with a higher PGS of CMSP, albeit at a borderline significant level. These results highlight the complex interactions between genetics and nutrition in sensitive developmental windows to shape lasting behavioural traits.

## Methods

### Sample

The Raine Study is a well-characterised pregnancy cohort located in Western Australia.[31–33] In brief, the recruitment of 2900 mothers during pregnancy began in 1989 from the King Edward Memorial Hospital and surrounding clinics. The Raine Study Gen1 and Gen2 antenatal data used in this paper included maternal questionnaire data from 18 weeks gestation and mother-baby dyads were followed through until birth. Subsequently, 2730 participating mothers (Generation 1/Gen1) and their 2868 offspring (Generation 2/Gen2) were followed up throughout childhood and adolescence with various phenotype assessments, including behaviour. The data for the present investigation were gathered from the Raine Study Gen2–1 year follow-up through to Gen2–17 year follow-up (the primary outcome being the longitudinal investigation of child behaviour in the Gen2 of the Raine Study at ages 2, 5, 8, 10, 14 and 17). For the present study, we used data from the sub-cohort that agreed to provide samples for genotyping. The study was conducted in accordance with the Declaration of Helsinki, and all participants provided written consent for their participation in the study at each follow-up. Ethics approvals were granted from the Human Research Ethics Committee of King Edward Memorial Hospital, Princess Margaret Hospital, the University of Western Australia, and the Health Department of Western Australia.

### Outcomes

The primary outcome was the total problem T-scores as assessed by the CBCL at ages 2, 5, 8, 10, 14 and 17 ($T-score_{TOT}$). In brief the CBCL is a clinically used psychometric instrument evaluating child behaviour wherein parents are asked to rate individual items (e.g. '*Acts too young for age*') on a 3-point Likert scale (0 = '*Not true*' 1 = '*Somewhat or sometimes true*', 2 = '*Very often or often true*'). Items are summed to generate a hierarchy of scores—the highest level is the total problem score summing all items; factor analysis has demonstrated clustering of items signifying externalising (aggressive and delinquent behaviour) and internalising (anxious/depressed, withdrawn and somatic complaints) problems (CBCL $T-score_{EXT}$ and CBCL $T-score_{INT}$).[34] The age 2 assessment was done with the preschool CBCL ages 2–3 years[35] and the age 5–17 assessment was done with the CBCL4/18.[27] Questionnaires with more than 8 missing items were discarded. Missing items for those with 8 or fewer missing items were treated as zero-scores. As a secondary outcome and to confirm the main findings, we used a teacher report form (TRF) filled out at age 10. This is a questionnaire by the same provider, which has parallel questions modified for the school context, and which follows a similar hierarchical construct order.

From each of the total internalising and externalising scales a T-score was derived (age and sex standardised), which is clinically used to group behaviours as normal (up to and including 59), borderline (60–64) or clinically significant problems (65+).

### Exposures

**Early nutrition.** We chose two primary exposures of interest: breastfeeding duration and maternal diet report at year 1.

Total breastfeeding duration was derived from maternal recall at follow-ups in years 1, 2 and 3 of life. Mothers were asked '*Did you breastfeed your baby?*' If they said '*no*' this was recorded as '*never breastfed*'. If they said '*yes*' they were asked '*At what age did you stop breastfeeding?*' with the answer recorded in months. Previous studies have found excellent reliability of maternal breastfeeding duration recall even up to 6 years after delivery.[36]

Diet was assessed using 24-h maternal dietary recall[37] and categorised with nutritionist supervision according to the categories of the Youth Healthy Eating Index that reflects dietary guideline adherence in children.[38] Parents were asked: '*please describe what food and drink your child has eaten in the past 24 h (please specify type of food/drink and quantity)*' and space was provided for breakfast, morning snack, lunch, afternoon snack, dinner and evening snack. Because of varying questionnaire responses related to portion size, only the food quality at each meal was included in a quasi-quantitative score, the eating assessment in toddlers (EAT) score.[37] In brief seven food sub-categories—wholegrain, vegetables, fruits, meat ratio ($\frac{white\ meat+egg+other\ protein\ sources}{red\ meat+processed\ meat}$), dairy, snack foods, and sweetened beverages—were scored 0–10 points depending on how many times they were offered at meals during the day. Our primary outcome, the total EAT score (range 0–70), summed these numbers, treating the first five components as positive and the latter two as negative (higher score proxying healthier diet). As a secondary measure to test the effects of dietary timing, a similar EAT score collected at age 3 years was included. In post hoc analysis, we also created three groups based on the food categories: plant-based (wholegrain, vegetables, and fruits), animal (meat ratio and dairy), and junk food products (snack foods and sweetened beverages). To facilitate effect size comparability for these categories, we used normalised scores.

**Genetics data and polygenic scores.** A total of 1593 participants had genotype data of approximately 560,000 single-nucleotide polymorphisms (SNPs) and ~95,000 copy number variants, obtained from an Illumina 660W Quad Array at the Centre for Applied Genomics, Toronto, Canada with quality control (QC) as per standard protocol. Plate controls and replicates with a higher proportion of missing data were excluded. We then assessed low genotyping success (>3% missing), excessive heterozygosity, gender discrepancies between the core- and genotyped data,

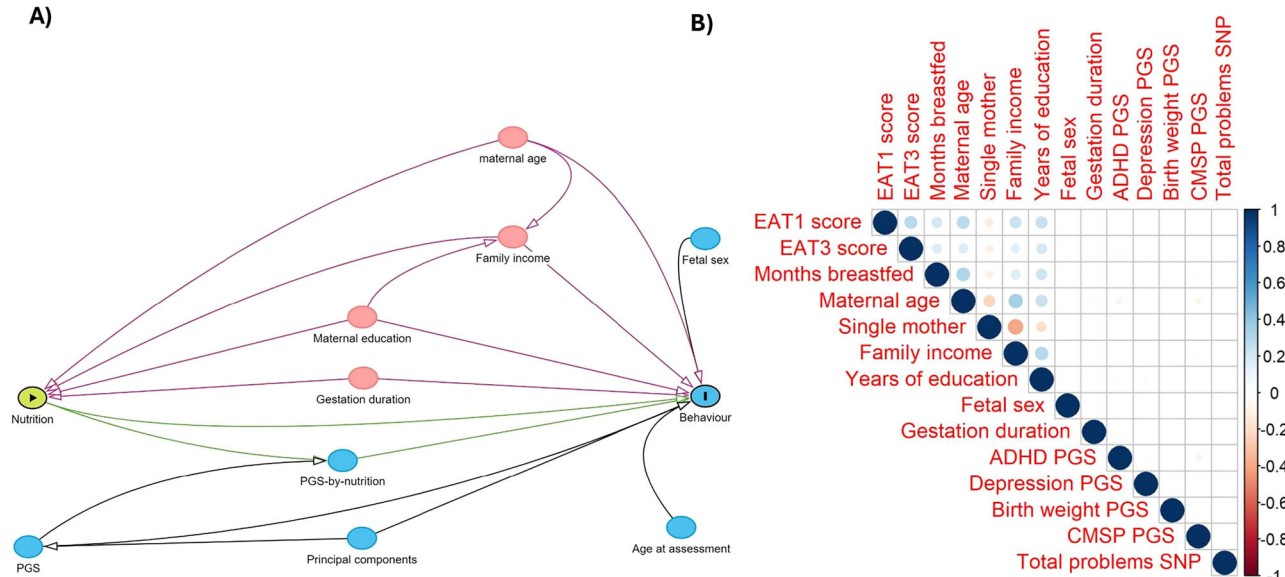

**Fig. 1 | The variables used for analysis. A** Conceptual framework of analysis and **B** correlation table of the variables used in regression. EAT eating assessment in toddlerhood, ADHD attention deficit hyperactivity disorder, CMSP chronic multisite pain, PGS polygenic score, SNP single nucleotide polymorphism.

and cryptic relatedness ($\pi > 0.1875$, in between second- and third-degree relatives). The SNP data were cleaned using plink[39] following the Wellcome Trust Case—Control Consortium protocol.[40] The exclusion criteria for SNPs included: Hardy–Weinberg-equilibrium $p < 5.7 \times 10^{-7}$; call-rate < 95%; minor-allele-frequency < 1%; and SNPs of possible strand ambiguity (i.e. A/T and C/G SNPs). The cleaned GWAS data were imputed using MACH software[41] across the 22-autosomes and X-chromosome against the 1000 Genome Project Phase I version 3.[42] After QC 1494 participants had SNPs and imputation resulted in 30,061,896 autosomal and 1,264,4493 x-linked SNPs. Principal components (PCs) analysis was then carried out, using SMARTPCA from v.3.0 of EIGENSOFT,[43] and PCs were generated to adjust for population stratification.

PGS were generated using the top hits from three GWASs likely to be associated with general psychopathology:[26] for ADHD we had 27 out of 27 available SNPs,[23] for depression, we had 95 out of 103,[24] and for CMSP we had 36 out of 39;[25] we had 146 out of 146 SNPs associated with own BW.[20,44] For the CBCL total problems GWAS, only one of two SNPs (rs10767094) was available, and the info score was borderline (0.30411).[21] Participant SNP data were extracted and recoded to correspond with increasing psychiatric risk (or increasing BW). It was then weighted using the beta-coefficients reported in the meta-analysis (for the depression GWAS, we used coefficients from the combined meta-analysis and 23andme replication), summed and re-scaled before the BW-PGS was calculated for each study participant. We confirmed the PGS predictive value for measures with relevant target data (adult self-report of depressive symptoms, a teacher assessment of ADHD, and the measured BW (data not shown)). To ease comparability in effect size, we used normalised scores.

**Confounders and additional variables.** Confounder selection for the relationship between early nutrition and child behaviour was based on an a priori search of the literature and visualised using directed acyclic graphs (Fig. 1A).[45] To avoid reverse causality (i.e. postnatal variables being a result of aberrant child behaviour), we used prenatal variables collected during weeks 16–18 of pregnancy. For the relationship between early nutrition and behaviour, we decided on a confounder set consisting of maternal age at birth, education level (highest year of finished education), family income (5 level variable treated as continuous) and civil status (dichotomised into single or partnered). Detailed descriptions of

gestation duration and questionnaire formulation can be found in the Supplementary Materials. Additionally, Gen2 participants age at assessment, biological sex collected at birth (one participant had missing data and sex was derived from a later questionnaire), and relevant PCs were included as fixed effects.

**Statistics.** Exposures, outcomes, and confounders were visually assessed and presented as summary statistics. We examined the Pearson correlation between variables, with particular focus on gene-environment correlations of nutritional measures and PGS that might bias the estimates.[45] The family income variables had more missing values than the remaining confounders ($n = 56$). To avoid bias and loss of power, these missing values were assigned the family income mean value for primary analysis (see also sensitivity analysis). The crude predictive value of the PGS and nutritional measures on T-score$_{TOT}$ was then regressed at each age.

Given the two primary exposures of interest (breastfeeding and year 1 diet) and the five genetic profiles with potential interaction, we pragmatically chose a significance threshold of 0.025 to limit chance findings while preserving power within our limited sample. The primary model using repeated CBCL T-scores was a linear mixed effects model with clustering at the participant ID level and varying intercepts. The interaction term between PGS and breastfeeding or year 1 EAT scores was then assessed. Model residuals showed some deviation from normality; therefore, we obtained two-tailed p-values from Z-statistics based on SE derived by non-parametric bootstrap (5000 resamples) to confirm primary results. Using Johnson–Neyman intervals, we derived a cut point to categorise those with and without significant effects from EAT scores. Results were reanalysed, assigning 'family income' missing values maximum and minimum values (instead of mean). Supplementary/sensitivity analyses of the model included a generalised logistic mixed effects model of a T-score$_{TOT}$ > 59, age 10 TRF T-score$_{TOT}$, specifically examining the age 2 rating, the exclusion of preterm births and using internalising and externalising outcomes. As we previously found sex-specific effects of the BW-PGS on behaviour[30] we also investigated sex-stratified BW-PGS-by-nutrition interaction models.

Post hoc we sought to identify the importance of specific food items and timing. Individual EAT-score categories (fibre, animal, and junk products) were added to the same model to examine the specificity of effects. Having found an effect of age-1 dietary fibre, we then made a final model

## Table 1 | Baseline demographics for the analytic sample (n = 1395)

|  | Breastfeeding (n = 1393) | EAT score year 1 (n = 1310) |
|---|---|---|
| **EAT score age 1 (points)** | | |
| Mean (SD) | 42.7 (±10.0) | 42.7 (±10.0) |
| Missing | 85 (6.1%) | 0 (0%) |
| **Breastfeeding duration (months)** | | |
| Mean (SD) | 8.0 (±7.2) | 8.1 (±7.1) |
| Missing | 0 (0%) | 2 (0.2%) |
| Biological sex (male) | 672 (48.2%) | 638 (48.7%) |
| ADHD PGS (SD) | 2.3 (±0.2) | 2.3 (±0.2) |
| Depression PGS (SD) | 2.0 (±0.1) | 2.0 (±0.1) |
| Birth weight PGS (SD) | 138.8 (±6.0) | 138.9 (±6.0) |
| Chronic multisite pain PGS (SD) | 0.5 (±0.1) | 0.5 (±0.1) |
| Total problems SNP present[a] | 911 (65.4%) | 863 (65.8%) |
| Maternal education (years) | 11.0 (±1.1) | 11.0 (±1.1) |
| **Family income[b]** | | |
| Mean (SD) | 3.9 (±1.1) | 3.9 (±1.1) |
| Missing | 56 (4.0%) | 51 (3.9%) |
| Civil status (partnered) | 1229 (88.2%) | 1164 (88.8%) |
| Maternal age (years) | 29.0 (±5.7) | 29.0 (±5.7) |
| Gestation duration (weeks) | 39.4 (±2.1) | 39.4 (±2.0) |
| Maternal ethnicity (caucasian) | 1353 (97.1%) | 1272 (97.0%) |
| **CBCL total problems age 2** | | |
| Mean (SD) | 51.0 (±9.0) | 51.1 (±9.0) |
| Missing | 295 (21.2%) | 275 (21.0%) |
| **CBCL total problems age 5** | | |
| Mean (SD) | 51.4 (±10.3) | 51.4 (±10.2) |
| Missing | 106 (7.6%) | 101 (7.7%) |
| **CBCL total problems age 8** | | |
| Mean (SD) | 49.8 (±11.1) | 49.8 (±11.1) |
| Missing | 126 (9.0%) | 114 (8.7%) |
| **CBCL total problems age 10** | | |
| Mean (SD) | 47.6 (±11.3) | 47.6 (±11.3) |
| Missing | 106 (7.6%) | 97 (7.4%) |
| **CBCL total problems age 14** | | |
| Mean (SD) | 46.7 (±11.5) | 46.8 (±11.6) |
| Missing | 133 (9.5%) | 120 (9.2%) |
| **CBCL total problems age 17** | | |
| Mean (SD) | 42.9 (±11.4) | 42.9 (±11.5) |
| Missing | 395 (28.4%) | 367 (28.0%) |

[a]Adenosin nucleotide at the SNP rs10767094.

[b]Family income had five levels: 1 less than $7000 2 $7,000–$11,999 3 $12,000–$23,999 4 $24,000–$35,000 5 $36,000 or more.

including both dietary fibre at ages 1 and 3. As the age 3 diet was collected after the first CBCL, only age 5–17 assessments were used for this analysis. For breastfeeding, we divided the sample into approximately equal portions by splitting the cohort from 0–2, 3–6, 7–12 and 13–18 months of breastfeeding to explore potential breastfeeding-sensitive age windows and potential non-linear effects.

## Results
### Sample description
We identified 1395 Gen2 participants with available data (1393 vs 1310 for analysis of breastfeeding duration and year 1 diet, respectively—see Supplementary Fig. 1 for flowchart). Participants had an average of 5.2 behavioural assessments; 12 were assessed only once, and 717 were assessed all six times. No clear differences were seen in variable distributions for participants used in breastfeeding and diet models (Table 1). Additional variables are described in the supplement (sT1). The analytic sample had more favourable maternal baseline demographics than the excluded Gen2 participants, assessed by birthweight and socioeconomic variables (sT2). The prespecified confounder variables correlated with both breastfeeding duration and diet, whereas the correlations for the PGSs were insignificant ($r < 0.1$) (Fig. 1). In isolation PGS did not predict total problems in childhood and adolescence; in contrast, longer breastfeeding and higher EAT1 scores were associated with fewer behaviour problems even when accounting for confounders (sT3).

### Polygenic score by nutrition interactions
Continuing to the primary question, we did not find statistically significant interactions between breastfeeding duration and PGSs at our predetermined significance threshold (all $p > 0.025$ see Table 2); however, a borderline result suggested that a higher genetic risk for CMSP amplified the positive effects of breastfeeding ($p = 0.03$) (Table 2). Exploring this borderline result, we split the cohort at the 0.025 alpha with a Johnson Neyman plot (Fig. 2) yielding 669 low- and 724 high-risk participants. This coincided with a pronounced effect difference between groups in models of externalising behaviour (Table 3, $B$: $-0.0816$, 95 % CI: [$-0.131$, $-0.0291$], $p = 0.002$), meaning the high-risk group saw a 0.124 point T-score$_{EXT}$ reduction for every 1-month increase in breastfeeding. The primary results were present in the term-born cohort and did not diminish with increasing age (sT4). In the pre-specified sex stratification, there was no signal to suggest interaction between BW-PGS and breastfeeding (sT5). Family income missing values did not appear to influence effect estimates (sT6). Post hoc we found that the differences in the primary linear mixed models coincided with lower T-score$_{TOT}$ for breastfeeding beyond 12 months in the genetic high-risk participants (Table 4).

For diet we found a significant interaction between ADHD-PGS and EAT1 score ($p = 0.0005$). Using a Johnson Neyman plot and splitting at the 0.025 alpha (Fig. 2), we defined two groups of 516 vs 794 participants with high and low ADHD-PGS, respectively. The remaining interactions did not reach significance; however, a signal suggested that the year-1 diet only benefitted males with a higher BW PGS in sex-stratified models (sT5).

In the ADHD low-risk group an improved diet—i.e. a higher EAT1-score—was associated with lower T-score$_{TOT}$ (Table 2, $B$: $-0.121$, 95% CI [$-0.171$, $-0.0704$]), whereas there was no effect in the high-PGS group. When examining the effects on lower-order behavioural domains we observed marginally larger effects on externalising than internalising behaviours (Table 3). In the supplementary analysis (sT4) there was a trend towards increasing diet effect size with advanced age. However this was not confirmed in the continuous model's three-way interaction, and the signal was already present at 2 years of age. Results were consistent in the term-only cohort and diminished but maintained directional similarity with an age-10 teacher assessment. The OR of meeting a clinically relevant cut point was also reduced with a higher EAT1 score. Family income missing values did not appear to influence effect estimates (sT6).

Post hoc we further explored the diet effects in the low-ADHD PGS group—first we split the components of the EAT score into three categories: plant-based, animal-based and junk food. We found that primarily plant-derived foods (Table 5, $B$: $-1.01$ per 1 SD, 95% CI: [$-1.50$, $-0.516$], $P$: <0.0001) drove the association with minimal effect modification if including year 3 plant-based food consumption ($B$: $-0.954$ per 1 SD, 95% CI: [$-1.44$ $-0.451$], $P$: 0.0002). We found a 30% effect size reduction for the age 3 plant-based consumption ($B$: $-0.664$, 95% CI: [$-1.20$, $-0.130$], $P$: 0.014). Breaking down the three components of the plant-based diet the effects were

**Table 2 | Interactions between genetic risk and early life nutrition for total behaviour problems**

| | ADHD PGS (27 SNPs) | Depression PGS (95 SNPs) | Chronic multisite pain PGS (36 SNPs) | Total problems * (1 SNP) | Birth weight PGS (146 SNPs) |
|---|---|---|---|---|---|
| **Breastfeeding N = 1393 O = 7191** | | | | | |
| Breastfeeding duration (0–38 months) | B: −0.0759 SE: 0.0267 CI: [−0.127, −0.0229] P: 0.004 | B: −0.0761 SE: 0.0262 CI: [−0.128, −0.0251] P: 0.004 | **B: −0.0801 SE: 0.0270 CI: [−0.133, −0.0268] P: 0.003** | B: −0.0496 SE: 0.0412 CI: [−0.128, 0.0336] P: 0.23 | B: −0.0759 SE: 0.0262 CI: [−0.128, −0.0252] P: 0.004 |
| Polygenic score (SD) | B: −0.177 SE: 0.269 CI: [−0.689, 0.366] P: 0.51 | B: 0.0427 SE: 0.274 CI: [−0.497, 0.575] P: 0.88 | **B: 0.633 SE: 0.292 CI: [0.0449, 1.19] P: 0.03** | B: 0.579 SE: 0.576 CI: [−0.517, 1.74] P: 0.31 | B: −0.0225 SE: 0.279 CI: [−0.579, 0.514] P: 0.94 |
| Interaction | B: 0.0324 SE: 0.0263 CI: [−0.0201, 0.0829] P: 0.90 | B: 0.000436 SE: 0.0263 CI: [−0.0505, 0.0525] P: 0.99 | **B: −0.0590 SE: 0.0276 CI: [−0.111, −0.00269] P: 0.03** | B: −0.0433 SE: 0.0508 CI: [−0.147, 0.0523] P: 0.39 | B: 0.0216 SE: 0.0247 CI: [−0.0265, 0.0703] P: 0.94 |
| **Year 1 diet N = 1310 O = 6780** | | | | | |
| EAT1-score (0–70 points) | **B: −0.0758 SE: 0.0193 CI: [−0.114, −0.0378] P: 0.00009** | B: −0.0748 SE: 0.0197 CI: [−0.113, −0.0358] P: 0.0001 | B: −0.0752 SE: 0.0198 CI: [−0.114, −0.0358] P: 0.0002 | B: −0.105 SE: 0.034 CI: [−0.171, −0.0379] P: 0.002 | B: −0.0729 SE: 0.0197 CI: [−0.111, −0.0342] P: 0.0002 |
| Polygenic score (SD) | **B: −2.69 SE: 0.800 CI: [−4.21, −1.07] P: 0.0008** | B: 0.956 SE: 0.936 CI: [−0.878, 2.79] P: 0.3 | B: 0.0130 SE: 0.72 CI: [−1.41, 1.41] P: 0.99 | B: −2.09 SE: 1.763 CI: [−5.54, 1.37] P: 0.24 | B: 1.26 SE: 0.831 CI: [−0.375, 2.88] P: 0.13 |
| Interaction | **B: 0.0625 SE: 0.018 CI: [0.0264, 0.0967] P: 0.0005** | B: −0.0198 SE: 0.0212 CI: [−0.0614, 0.0218] P: 0.35 | B: 0.00498 SE: 0.0162 CI: [−0.0266, 0.0369] P: 0.76 | B: 0.0456 SE: 0.0398 CI: [−0.0330, 0.123] P: 0.25 | B: −0.0259 SE: 0.0189 CI: [−0.0629, 0.0113] P: 0.17 |

Primary models. Estimates are from models including an interaction term between nutrition and PGS. All models adjusted for sex, age at assessment, maternal age at birth, maternal education length, family income, gestational duration, civil status and principal components. Confidence interval (CI) and p-value in mixed models based on bootstrapped Standard Errors (SE) and two-sided z-test. *Estimates signify the presence of the risk allele. **Bold:** Significant or borderline significant interactions.

largest for wholegrain, then fruit and least for vegetables (sT7). No significant associations emerged with age 3 plant-based foods.

## Discussion

Using longitudinal data throughout childhood and adolescence, we have demonstrated the interplay between early nutrition and psychiatric PGS in determining behaviour. Consistent with our a priori hypothesis, a borderline signal suggested that individuals with a high genetic risk for CMSP had larger benefits from longer breastfeeding; however, contrary to our a priori hypothesis, a strong signal suggested that individuals with a low ADHD risk benefitted from improved diet in the first year of life. We note that the conventionally viewed 'good' early life nutrition (i.e. higher EAT score and longer duration of breastfeeding) was not associated with worse outcomes in any genetic subgroup. Nevertheless, contingency on genetic risk was considerable.

For diet, the ADHD low-risk group saw a 0.121 point lower T-score$_{TOT}$ per 1 point higher EAT-score (scale 0–70). This association would, in a causal framework, translate to an 8–9 points improvement (SD of T-score$_{TOT}$ = 10 points) from the worst to the best diet, whereas the high-risk group would translate to only a 0–1 point improvement. The behaviour association was detectable at age two, was consistent across ages with no sign of effect size reduction with increasing age, and we saw non-significant but directionally similar results with the teacher assessments. The higher EAT score also reduced the 'borderline' problems category (T-score > 60), hinting at clinical significance. We did not see pronounced differences for specific behavioural patterns, but the effect sizes were larger for externalising compared to internalising behaviour.

For breastfeeding, we highlight that the p-value of the primary interaction did not cross the pre-specified significance threshold. Nevertheless, a signal suggested that every month of breastfeeding was associated with 0.11

points lower T-score$_{TOT}$ in those with a higher genetic risk for CMSP. The WHO recommends 2 years of breastfeeding and, in a causal framework, going from 0 to 24 months. This would translate to a 2–3 point reduction in T-score$_{TOT}$, whereas the low-risk group would see around a 1 point decrease. Exploring brackets of breastfeeding in the high-risk group, most breastfeeding benefits in the high-risk group were seen after 6 months, whereas the low-risk group saw no additional T-score$_{TOT}$ reductions beyond two months. Exploring breastfeeding further models of T-score$_{EXT}$ showed a strong interaction hinting at behaviour-specific effects. This is consistent with the increased hostility seen in the 'never-breastfed' group of the Young Finns study on a total population level.[46] The interaction was robust across ages and in those born at term.

The mechanisms underlying our associations are not clear. In psychiatry the diathesis-stress model proposes that vulnerability interplays with environmental stressors to provoke mental illness.[3] We believe that the borderline interaction between CMSP and breastfeeding is consistent with this theory, as shorter breastfeeding could be considered a stressor.[47] From a mechanistic point of view breastfeeding has been proposed to alter neurodevelopment through several mechanisms. The long-chain fatty acids in breast milk have been proposed to enhance white matter myelination in the first 2 years of life.[48] Alternatively changes in the gut microbiome from breastfeeding could affect the gut-brain axis leading to behavioural changes. Finally increased breastfeeding could enhance maternal bonding and affect both child behaviour and maternal evaluation of child behaviour. The first year of life has been considered critical for microbiome establishment[49] and white matter development.[50] As breastfeeding duration up to 2 years influences maternal sensitivity[51] and the associations in the present study differed primarily beyond 12 months, we propose that the effects relate to increased effects of child-maternal bonding in psychiatrically vulnerable offspring.

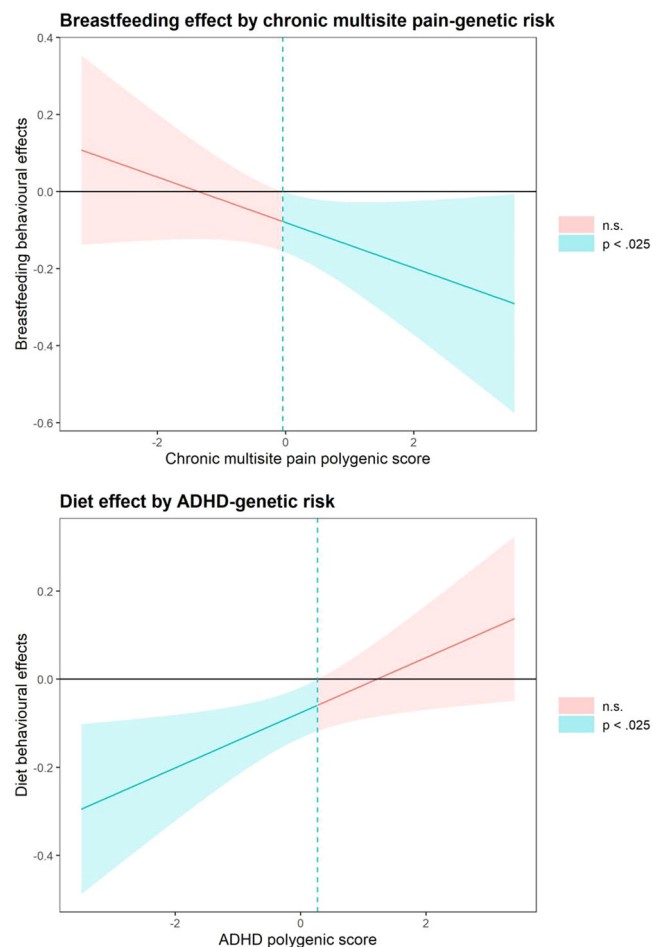

**Fig. 2 | Splitting the cohort by genetic risk.** We see that the effects of age 1 diet on child behaviour total problems diminishes with increasing ADHD polygenic score, whilst the effect of breastfeeding increases with increasing chronic multisite pain score. Error bands signify the 97.5% confidence interval.

## Table 3 | Behavioural specificity of nutrition association

| | Total problems<br>Diet effects by genetic ADHD risk | Externalising | Internalising |
|---|---|---|---|
| B EAT-score age 1 (points)<br>Low genetic risk<br>($n = 794$ $O = 4114$) | B: −0.121<br>SE: 0.0258<br>CI: [−0.171, −0.0704]<br>P: 0.000003 | B: −0.103<br>SE: 0.0244<br>CI: [−0.150, −0.0541]<br>P: 0.00002 | B: −0.0861<br>SE: 0.0229<br>CI: [−0.131, −0.0410]<br>P: 0.0002 |
| B EAT-score age 1 (points)<br>High genetic risk<br>($n = 516$ $O = 2666$) | B: −0.0116<br>SE: 0.0302<br>CI: [−0.0718, 0.0465]<br>P: 0.7 | B: 0.00148<br>SE: 0.0281<br>CI: [−0.0539, 0.0564]<br>P: 0.96 | B: −0.0242<br>SE: 0.0276<br>CI: [−0.0799, 0.0285]<br>P: 0.38 |
| Interaction: EAT score by continuous ADHD-PGS<br>($N = 1310$ $O = 6780$) | Primary outcome—See Table 2 | B: 0.0560<br>SE: 0.0163<br>CI: [0.0236, 0.0876]<br>P: 0.0006 | B: 0.0392<br>SE: 0.0158<br>CI: [0.00676, 0.0688]<br>P: 0.013 |
| Breastfeeding effects by genetic chronic multisite pain risk | | | |
| Breastfeeding (months)<br>Low genetic risk<br>($n = 669$, $o = 3458$) | B: −0.0486<br>SE: 0.0355<br>CI: [−0.120, 0.0193]<br>P: 0.17 | B: −0.0259<br>SE: 0.0335<br>CI: [−0.0921, 0.0390]<br>P: 0.44 | B: −0.0604<br>SE: 0.0321<br>CI: [−0.124, 0.00174]<br>P: 0.02 |
| Breastfeeding (months)<br>High genetic risk<br>($n = 724$ $o = 3733$) | B: −0.108<br>SE: 0.0397<br>CI: [−0.184, −0.0289]<br>P: 0.007 | B: −0.124<br>SE: 0.0380<br>CI: [−0.197, −0.0476]<br>P: 0.001 | B: −0.0770<br>SE: 0.0350<br>CI: [−0.144, −0.00710]<br>P: 0.028 |
| Interaction: breastfeeding by continuous CMSP-PGS<br>($n = 1393$ $O = 7191$) | Primary outcome—See Table 2 | B: −0.0816<br>SE: 0.0260<br>CI: [−0.131, −0.0291]<br>P: 0.002 | B: −0.0192<br>SE: 0.0236<br>CI: [−0.0637, 0.0288]<br>P: 0.42 |

Exploring different behaviours following cohort stratification. All models adjusted for sex, age at assessment, maternal age at birth, maternal education length, family income, gestational duration, civil status and principal components. Confidence interval (CI) and *p*-value in mixed models based on bootstrapped standard errors (SE) and two-sided *z*-test.

The diet-by-PGS results in ADHD conflict with the diathesis-stress model and we see two potential explanations for this surprising result. First the genetics of ADHD may determine behaviour to such an extent that high-risk individuals are not amenable to influences from early environmental exposures such as diet. Indeed, the heritability of ADHD has been estimated at 70–80%[23] leaving little room for the environment; however, our PGS was a poor predictor in and of itself and only showed a significant prediction of T-score$_{TOT}$ at age 2 years. As such our instrument is likely insufficient to capture such an exposure-resistant genetic predisposition. Alternatively we observed an age two increase in behaviour problems associated with the ADHD-PGS (sT3), which opens the possibility that the interaction stems from gene-environment correlation.[52] i.e. if the early increase in problem behaviour manifested as a refusal to eat less palatable foods, only those offered the 'healthy' foods and with a lower genetic risk would eat them. From a mechanistic perspective, we found evidence that plant-based especially wholegrain, foods at age 1 exerted strong effects. We postulate that dietary fibre is the more important constituent underlying the association (as compared to polyphenols or vitamins more abundant in fruits and vegetables). Consistent with this, year 1 of life is considered 'crucial'[53] in establishing a healthy gut microbiome and the robustness of the year 1 estimate to including year 3 diet hints at an early window of opportunity for optimising diet. Supporting this notion, differences in gut-microbiome composition in adult ADHD have been shown in many studies.[54] Alternatively short-chain fatty acids generated from gut-microbe metabolism of dietary fibre could enhance white matter myelination, as explained above.

Our study had several strengths. The key strength was the long-term follow-up of the same cohort over many years with repeated assessments throughout childhood and adolescence. A further strength was the use of consistent validated questionnaires, which were completed by multiple raters. The repeated follow-up in early life allowed us to conduct analyses regarding the time sensitivity of nutritional exposures and minimise the recall bias risk. Collection of confounders before birth ensured that we were not adjusting for downstream effects of offspring behaviour, and the prospective nature should minimise the risk of reverse causality. In addition the effects of the age-1 diet were robust and largely unaltered even when the age-3 measures were included, suggesting that the association is not explained by the correlation of early and late diet.

The study also has some limitations. The Raine Study is from 1989; our results are not certain to generalise to later cohorts—i.e. modern dietary and breastfeeding patterns could change the GxE interaction in a more contemporary setting; furthermore, the diet exposure was a 24-h recall and could be an imprecise measure of general diet-patterns, which would bias our results towards the null. Another limitation is that our hypothesis-driven approach to PGS selection is unlikely to have picked the most influential PGS. A broad screening of genetic instruments would require stringent significance thresholds and given our limited sample size, we refrained from this. Our results should also be interpreted with caution as our *p*-value threshold for significance was pragmatically chosen to account for a limited sample size, rather than a strict Bonferroni correction. Although we had a good follow-up of the genotyped cohort in the Raine Study (93 % of the 1494 genotyped

**Table 4 | Breastfeeding brackets**

| | 0−1 months (intercept) N = 260 | 2−5 months N = 358 | 6−11 months N = 412 | 12−38 months N = 363 |
|---|---|---|---|---|
| Breastfeeding quartiles Low genetic risk (N = 669, O = 3458) | B: 67.6 SE: 3.3524 CI: [60.8, 73.9] | B: −1.00 SE: 0.8386 CI: [−2.65, 0.641] P: 0.23 | B: −1.59 SE: 0.7706 CI: [−3.11, −0.0881] P: 0.04 | B: −1.14 SE: 0.8316 CI: [−2.79, 0.473] P: 0.17 |
| Breastfeeding quartiles High genetic risk (N = 724, O = 3733) | B: 72.2 SE: 3.0577 CI: [66.0, 78.0] | B: 0.223 SE: 0.8256 CI: [−1.42, 1.82] P: 0.79 | B: −1.26 SE: 0.8267 CI: [−2.90, 0.341] P: 0.13 | B: −2.55 SE: 0.8560 CI: [−4.23 −0.875] P: 0.003 |
| Interaction: breastfeeding by continuous CMSP-PGS (N = 1393 O = 7191) | B: 70.2 SE: 2.1561 CI: [65.8, 74.3] | B: 0.490 SE: 0.6196 CI: [−0.738, 1.69] P: 0.43 | B: 0.0980 SE: 0.6092 CI: [−1.09, 1.29] P: 0.87 | B: −1.47 SE: 0.6207 CI: [−2.68, −0.248] P: 0.018 |

Exploring the timing of breastfeeding. All models adjusted for sex, age at assessment, maternal age at birth, maternal education length, family income, gestational duration, civil status and principal components. Confidence interval (CI) and p-value in mixed models based on bootstrapped standard errors (SE) and two-sided z-test.

**Table 5 | Exploring diet**

| | Higher ADHD PGS N = 516, O = 2666 | Lower ADHD PGS N = 794, O = 4114 | Lower ADHD PGS including age 3 diet[b] N = 791, O = 3489 |
|---|---|---|---|
| Plant based age 1[a] | B: −0.00685 SE: 0.2927 CI: [−0.583, 0.565] P: 0.98 | B: −1.01 SE: 0.2517 CI: [−1.50, −0.516] P: 0.00006 | B: −0.938 SE: 0.2746 CI: [−1.47, −0.398] P: 0.0006 |
| Junk food age 1[a] | B: −0.570 SE: 0.2846 CI: [−1.13, −0.0111] P: 0.05 | B: −0.253 SE: 0.2514 CI: [−0.749, 0.236] P: 0.31 | NA |
| Animal derived age 1[a] | B: 0.250 SE: 0.3153 CI: [−0.388, 0.848] P: 0.43 | B: −0.461 SE: 0.2390 CI: [−0.922, 0.0148] P: 0.05 | NA |
| Plant based age 3[a] | NA | NA | B: −0.664 SE: 0.2718 CI: [−1.20, −0.130] P: 0.014 |

Exploring individual components and time sensitivity of diet effects. All models adjusted for sex, age at assessment, maternal age at birth, maternal education length, family income, gestational duration, civil status and principal components. Confidence interval (CI) and p-value in mixed models based on bootstrapped standard errors (SE) and two-sided z-test.
[a]To ease group comparison, the food scores were standardised (subtracting mean dividing by SD).
[b]To minimise reverse causality for the age three dietary measures, we excluded the age two total problem score, using only ages 5–17.

participants), the need for genetic information introduced selection bias and our sample consisted of half (51.3 %) of the original cohort. This could limit the generalisability of our exposure-outcome relationships; however, we did not see signs of selection bias in the PGS histograms and previous studies have suggested that exposure-outcome relationships in the Raine Study follow-ups vs dropouts are similar.[33] We had excellent data quality for potential confounders; however, potential residual confounding limits any causal conclusions regarding nutrition effect estimates.[45] In contrast the underlying confounder architecture would have to differ between high- and low-risk groups to explain away the interaction estimates as the genes are subject to random allocation.

Future basic research should be directed at understanding the impact of timing for early nutritional effects on the gut microbiome; furthermore, the impact of ADHD risk on early-life eating habits is warranted, given the surprising lack of diet-behaviour association in the high-risk ADHD-PGS group. For breastfeeding, our results need replication before drawing robust conclusions. The detectability of effects at age 2 years is encouraging for future early nutrition RCTs, and we hope interventionists studying early nutrition will include genetic profiling in trials for subgroup analysis.

In conclusion, we show for the first time that early nutrition and psychiatric genetic risk interact in shaping behaviour throughout childhood and adolescence.

## Data availability
The individual-level data from the Raine Study cannot be made available or stored on another, non-Raine Study public repository due to legal and ethical constraints. The Raine Study is a publicly available data provider and data can be accessed by interested researchers upon reasonable request. To request access, please contact the Raine Study team at rainestudyscience@uwa.edu.au.

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

## Acknowledgements
We would like to acknowledge the Raine Study participants and their families for their ongoing participation in the study and the Raine Study team for study co-ordination and data collection. We also thank the NHMRC and the Raine Medical Research Foundation for their long-term contribution to funding the study over the last 30 years. The core management of the Raine Study is funded by The University of Western Australia, Curtin University, The Kids Research Institute Australia, Women and Infants Research Foundation, Edith Cowan University, Murdoch University and The University of Notre Dame Australia. The Pawsey Supercomputing Centre provided computation resources to carry out analyses required with funding from the Australian Government and the Government of Western Australia. We acknowledge the Raine Medical Research Foundation, the NHMRC (Sly et al., ID 211912; Stanley et al., ID 003209; Stanley et al., ID 353514; Palmer et al., ID 572613; Beilin et al., ID 403981; Huang et al., ID 1059711) and Canadian Institutes of Health Research-CIHR (Lye et al., MOP-82893) for funding of the Gen1 antenatal data collection and the Gen2 data collection from ages 1–17, including genetic data.

## Author contributions
Lars Meinertz Byg: conceptualisation, methodology, formal analysis, writing-original draft, writing-review and editing; Carol A Wang: methodology, writing-review and editing; John Attia: methodology, writing-review and editing; Andrew J.O. Whitehouse: resources, writing-review and editing; Wendy Oddy: resources, writing-review and editing; Jonathan Hirst: writing-review and editing, supervision, project administration; Craig E Pennell: conceptualisation, resources, writing-review and editing, supervision, project administration, funding acquisition.

## Competing interests
The authors declare no competing interests.
