## [Transparent Peer Review File · Communications Medicine]

Nutrition in Early Life Interacts With Genetic Risk to Influence Preadult Behaviour in the Raine Study.

Corresponding Author: Professor Craig Pennell

Version 0:

Reviewer comments:

Reviewer #1

(Remarks to the Author)

This is an interesting study of possible interactions between early childhood diet and genetic risk of psychiatric disorders in relation to child and adolescent behaviour problems. It uses approximately half of the offspring studied in the important Australian Raine longitudinal study. The paper does find some significant results, but this reviewer found the study design difficult to follow. Nevertheless with extensive clarification this will be an important contribution to the literature.

Suggested alterations

1. The title is confusing and doesn't really convey the content or aims of the paper.

2. The Abstract:

a) Diet is measured using an EAT score based on a 24 hour recall at age 2, but there is no indication as to what is being measured, or whether an increase in the score indicates whether the diet is improved in a specific way or not.

b) Please state that the CBCL score used is the sum of the scores at the different ages.

c) The first sentence of the Conclusion implies that you have a causal relationship, but you have only shown an association. I suggest that you put more caveats into this section and, for example, suggest that the results would carry more weight if other longitudinal studies had similar results.

d) Also, in regard to the above sentence, the implication is that the effect is only on behaviour in the first year – but surely the effect is of diet in the first year on behaviour throughout the period. Please clarify.

3. The Introduction

a) Reference 5 is quoted as though it refers to childhood behaviour, but it is reporting on adults.

b) Please check the validity of all other references. For example, the first paragraph is about psychiatric disorders in general but some of the references are only relevant to specific disorders – e.g. depression [2], ADHD [5].

4. The Results

a) Tables 1 and 2: the EAT score should be in caps not lower case.

b) Lines 210, 214: T2 and T3 should be substituted with Table 2 and Table 3 respectively.

5. The Discussion

a) Lines 292-3: you state 'given the strong significance it is unlikely to be spurious'. This statement needs to be substantiated with a test to determine the likelihood of a hidden confounder, such as the E-risk.

b) Lines 318-319: you state that selection bias is minimised because 93% of the genotyped children were followed up. This only refers to the genotyped, but the non-genotyped are likely to provide a much more convincing source of bias. True you mention this later, but it would be better to combine both sentences together.

Reviewer #2

(Remarks to the Author)

Abstract

Please give an indicator of the importance/scale of effect of '1 EAT1 unit' in abstract to gauge strength of effect.

Phrasing of the conclusion that 'Post hoc analysis suggested that plant-based food at age one drove the association' is not clear – did this refer to variety, frequency, or amount consumed? Please specify the unit of measurement.

Abstract states breastfeeding duration was ascertained prospectively but in the methods it describes recall at 1, 2 and 3 years.

Because of the lack of clarity about the timing and duration of the dietary effects on behavioural outcomes, I question whether the abstract's concluding sentence is correctly phrased. 'Nutrition and psychiatric genetic risk interact to determine child behaviour in the first year of life' – should this be 'Nutrition and psychiatric genetic risk interact in the first year of life to determine child behaviour'?

How confident are the authors of this conclusion: 'Breastfeeding and early diet can potentially reduce behaviour problems in later life'. It was an observational study. Perhaps more difficult children are harder to get to eat a healthy diet or maintain longer breastfeeding?

Introduction:

Unclear relevance of the following statement ; "albeit the most recent efforts yielded two hits [19]" – please expand/explain or remove.

Please give direction of effects in the following statement: "Surprisingly, an association has also emerged between birth weight (BW) PGS and childhood behaviour [26-28] with different effects in males and females [29]"

The rationale for year 1 diet as an exposure requires greater justification. There is insufficient literature in the introduction which explains why this measure, or its timing, was important to examine. What is meant by year 1 diet – overall diet quality? Types of foods used in complementary feeding? How is 'good nutrition' defined?

The rationale for the direction of the hypothesised interactive nature of PGS and nutrition needs stronger justification. Why was it hypothesised that breastfeeding and 'good nutrition' would have a greater beneficial effect on problem behaviour in those at greater genetic risk? It is equally plausible that positive effects of nutrition would be observed but be less successful in mitigating problems for those at greater genetic risk. Is there other justification for the direction of this prediction that can be included here?

Method

Breastfeeding measurement: what was the specific question asked of mothers? Duration, exclusivity? Introduction of alternative milks? Age of cessation of any breastfeeding? Given the dose dependent relationships observed with regard to positive effects of breastfeeding in other domains, this information is critical.

Presumably diet was assessed at 1 year. Has the EAT score been well validated against other measures of dietary intake or quality or widely used in dietary assessments of 1 year olds? Are higher scores indicative of better or worse diet quality?

It was not clear what was meant by the following statement:

"Because of varying questionnaire responses, only the food quality was included in a quasi-quantitative score". Please clarify.

Statistics: $p < .025$ as a Bonferroni correction is only accurate if there are only two outcomes (analyses) rather than only two primary exposures.

Why were breastfeeding data split into groups rather than retaining power by keeping the duration as a continuous variable?

Why were effects of breastfeeding and diet quality tested separately? It is plausible there is an additive effect of these.

How was the repeated nature of the outcome measure treated within the analysis?

What is figure 1 b intended to show?.

It is not clear at all what the duration or longevity of the effects you have documented are because it is not clear how the behaviour outcome is treated in the analysis. If the effects are observed at 17 this is very different from observations at 2 years. Does the dietary influence occur early, triggering positive developmental cascades? Or does it manifest later, implying other mechanistic effects? Whether there were important effects that determined the difference between clinically significant problems or not was also not clearly presented.

The results should present all analyses and whilst some may be presented in supplementary materials, the reader should be able to access all of the results. Currently the manuscript feels selective in its presentation, especially given the number of exploratory and post hoc analyses.

Version 1:

Reviewer comments:

Reviewer #1

(Remarks to the Author)

Reviewer #2

(Remarks to the Author)

The authors have addressed most of my concerns. However, I still feel the answer to the following is unsatisfactory:

Original Query (Abstract): How confident are the authors of this conclusion: 'Breastfeeding and early diet can potentially reduce behaviour problems in later life'. It was an observational study. Perhaps more difficult children are harder to get to eat a healthy diet or maintain longer breastfeeding?

Action: We have now changed the wording to "Targeted breastfeeding and early diet interventions have the potential reduce later behaviour problems.(Page 2, Line 23-24).

This conclusion is still the same. I would recommend a re-wording or addition to the sentence that highlights the need for a trial to test this potential - indeed the conclusions within the discussion itself are more appropriately tempered.

Typo in abstract: 'a strong signal suggested that 1 EAT1 point increase result in a reduced..' result should read resulted.

General comment

We thank the reviewers for their time and thought. The manuscript has certainly improved in clarity and readability following their comments.

Format:

We have kept our comments in red font, with division of our **response** and **action** when relevant. Citations are kept italic and in “brackets” with reference to the clean manuscript page and line number.

Reviewer #1 (Remarks to the Author):

This is an interesting study of possible interactions between early childhood diet and genetic risk of psychiatric disorders in relation to child and adolescent behaviour problems. It uses approximately half of the offspring studied in the important Australian Raine longitudinal study, The paper does find some significant results, but this reviewer found the study design difficult to follow. Nevertheless with extensive clarification this will be an important contribution to the literature.

Suggested alterations

1. The title is confusing and doesn't really convey the content or aims of the paper.

Action: We have changed the title to “*Nutrition in Early Life Interacts with Genetic Risk to Influence Preadult Behaviour: Findings from the Raine Study.*” (Page1, Line 1-3)

2. The Abstract:

a) Diet is measured using an EAT score based on a 24 hour recall at age 2, but there is no indication as to what is being measured, or whether an increase in the score indicates whether the diet is improved in a specific way or not.

Action: We have now changed the wording to “*..and toddler diet derived by an age-one 24-hour maternal recall (EAT1, scale 0-70, SD 10, higher scores proxying healthy diet).*” (Page 2, Line 9-10)

b) Please state that the CBCL score used is the sum of the scores at the different ages.

Action: We have now changed the wording to “*In confounder-adjusted mixed-effects models of CBCL_{TOT} throughout follow-up we examined nutrition-by-PGS interactions*” (Page 2, Line 12-13)

c) The first sentence of the Conclusion implies that you have a causal relationship, but you have only shown an association. I suggest that you put more caveats into this section and, for example, suggest that the results would carry more weight if other longitudinal studies

had similar results.

Action: We have now modified the causal language and additionally changed the wording to “*Contrary to our hypothesis, we saw dietary benefits in individuals with lower ADHD PGS, necessitating replication*” (Page 2, Line 22-23)

d) Also, in regard to the above sentence, the implication is that the effect is only on behaviour in the first year – but surely the effect is of diet in the first year on behaviour throughout the period. Please clarify.

Action: We have now changed the wording to “*Nutrition in early life and psychiatric genetic risk may interact to determine child behaviour.*” (Page 2, Line 22)

3. The Introduction

a) Reference 5 is quoted as though it refers to childhood behaviour, but it is reporting on adults.

b) Please check the validity of all other references. For example, the first paragraph is about psychiatric disorders in general but some of the references are only relevant to specific disorders – e.g. depression [2], ADHD [5].

Response: We apologise for the imprecisions and have been over the references and added/Corrected as appropriate.

4. The Results

a) Tables 1 and 2: the EAT score should be in caps not lower case.

Action: This has been amended

b) Lines 210, 214: T2 and T3 should be substituted with Table 2 and Table 3 respectively.

Action: This has been amended

5. The Discussion

a) Lines 292-3: you state ‘given the strong significance it is unlikely to be spurious’. This statement needs to be substantiated with a test to determine the likelihood of a hidden confounder, such as the E-risk.

Action: The reviewer is correct, and the questionable part has been removed. The sentence now reads “*The diet-by-PGS results in ADHD conflict with the diathesis-stress model and we see two potential explanations for this surprising result.*” (Page 13, line13-14)

b) Lines 318-319: you state that selection bias is minimised because 93% of the genotyped children were followed up. This only refers to the genotyped, but the non-genotyped are likely to provide a much more convincing source of bias. True you mention this later, but it would be better to combine both sentences together.

Action: This part of the discussion has now been combined and suited in the “limitations” section. It reads: “Although, we had a good follow-up of the genotyped cohort in the Raine Study (93 % of the 1494 genotyped participants), the need for genetic information introduced selection bias and our sample consisted of half (51.3 %) of the original cohort. This could limit the generalisability ..” (Page 14, Line 19-21)

Reviewer #2 (Remarks to the Author):

Abstract

Please give an indicator of the importance/scale of effect of ‘1 EAT1 unit’ in abstract to gauge strength of effect.

Action: This was also remarked by R1 and the abstract now reads: “..and toddler diet derived by an age-one 24-hour maternal recall (EAT1, scale 0-70, SD 10, higher scores proxying healthy diet).” (Page 2, Line 9-10)

Phrasing of the conclusion that ‘Post hoc analysis suggested that plant-based food at age one drove the association’ is not clear – did this refer to variety, frequency, or amount consumed?

Action: This sentence now reads: “Post hoc analysis suggested that plant-based food consumption drove the favourable EAT1-CBCL_{TOT} association.” (Page 2, Line 19-20)

Please specify the unit of measurement.

Response: The unit of measurement is the point-score. We have tried further specifying the score-characteristics as per reviewer request and hope the current description is sufficient. The full description of the score is found in the methods-section (Page 6, Line 14-28)

Abstract states breastfeeding duration was ascertained prospectively but in the methods it describes recall at 1, 2 and 3 years.

Action: We have now changed the wording to “Breastfeeding duration was recalled at age 1, 2 and 3 follow-up,” (page 2, line 8-9)

Because of the lack of clarity about the timing and duration of the dietary effects on behavioural outcomes, I question whether the abstract’s concluding sentence is correctly phrased. ‘Nutrition and psychiatric genetic risk interact to determine child behaviour in the first year of life’ – should this be ‘Nutrition and psychiatric genetic risk interact in the first year of life to determine child behaviour’?

Action: We have now changed the wording to “Nutrition in early life and psychiatric genetic risk may interact to determine child behaviour.” (Page 2, Line 22)

How confident are the authors of this conclusion: ‘Breastfeeding and early diet can potentially reduce behaviour problems in later life’. It was an observational study. Perhaps more difficult children are harder to get to eat a healthy diet or maintain longer breastfeeding?

Action: We have now changed the wording to “Targeted breastfeeding and early diet interventions have the potential reduce later behaviour problems.” (Page 2, Line 23-24)

Introduction:

Unclear relevance of the following statement ; “albeit the most recent efforts yielded two hits [19]” – please expand/explain or remove.

Action: We have now changed the wording to “*albeit the most recent efforts yielded two significant genetic variants associated with total problems [21];*” (Page 3, Line 23-24)

Please give direction of effects in the following statement: “Surprisingly, an association has also emerged between birth weight (BW) PGS and childhood behaviour [26-28] with different effects in males and females [29]”

Response: We also agree that specificity is preferable and from a phenotype standpoint the association is uniformly that lower BW is associated with more psychiatric problems; however, for the genotype the direction is a little difficult to give because the effects depend on the population, the conditioning on the maternal genotype and inclusion of BW-phenotype. In the paper by Orri et al they found that decreased birthweight was associated with increased adverse outcomes (e.g. ADHD, suicide attempts and lower intelligence). In the D’Urso paper the effects were more heterogeneous and subtle and dependent on genotype. Finally, our own work (cited 30) has suggested that the long term effects of birthweight arise in an interplay between phenotype and genotype with different effects in males and females – notably a monotonic positive association between BW-PGS in females with aggressive/ADHD problems and a phenotype dependent effect in males (frail male). I.e. exceeding or failing to meet the genetically determined growth potential. Inclusion of such complexity is however beyond the scope of the current paper which seeks to explore the interaction between genetics and early diet. Nevertheless, as we had demonstrated psychiatric effects of the BW/PGS and previously demonstrated an interaction between BW-PGS and early diet in predicting cardiometabolic outcomes it seemed intuitive to include the score in our analysis. This was included in the final part of the sentence “*Surprisingly, another association has emerged between between birth weight (BW) PGS and childhood behaviour [28-30] with different effects in males and females [30] and our group previously demonstrated an interaction between BW-PGS and early nutrition in predicting adult cardiometabolic risk [20].*” (page 3, line 30 and page 4, line 1-3)

The rationale for year 1 diet as an exposure requires greater justification. There is insufficient literature in the introduction which explains why this measure, or its timing, was important to examine. What is meant by year 1 diet – overall diet quality? Types of foods used in complementary feeding? How is ‘good nutrition’ defined?

Response: We used the year 1 diet because this was the earliest diet measure, we had, working from the hypothesis of increased early plasticity. “Good nutrition” was defined by the dietitians who constructed the EAT score. This would mean higher reporting of year 1 wholegrain, vegetables, fruits, meat ratio ((white meat+ egg+ other protein sources)/(red meat + processed meat)) and lower reporting of year 1 snack foods, and sweetened beverages. We agree with the reviewer that “Good nutrition” could be considered contentious. For this very reason we split the categories in post hoc analysis to probe what specific food groups drove the benefits.

Action: We have rephrased the final paragraph to say “*The two exposures of interest (breastfeeding and early diet) were therefore screened for interaction with a final of five*

polygenic scores to predict behaviour problems in childhood and adolescence. We hypothesised that breastfeeding and nutrition effects on problem behaviour would be larger in genetically vulnerable individuals.” (page 4, line 5-7)

The rationale for the direction of the hypothesised interactive nature of PGS and nutrition needs stronger justification. Why was it hypothesised that breastfeeding and ‘good nutrition’ would have a greater beneficial effect on problem behaviour in those at greater genetic risk? It is equally plausible that positive effects of nutrition would be observed but be less successful in mitigating problems for those at greater genetic risk. Is there other justification for the direction of this prediction that can be included here?

Response: We based this on the widely accepted “diathesis-stress” model of psychiatric illnesses. In essence it deals with psychiatric vulnerability and stressor load. Perhaps naively, we *a priori* thought of poor early diet as a stressor akin to any other. This turned out to be incorrect as we discuss later in the manuscript.

Action: For clarity we have tried rephrasing the hypothesis “*We hypothesised that breastfeeding and nutrition effects on problem behaviour would be larger in genetically vulnerable individuals.*” (page 4, line 6-7)

Method

Breastfeeding measurement: what was the specific question asked of mothers? Duration, exclusivity? Introduction of alternative milks? Age of cessation of any breastfeeding? Given the dose dependent relationships observed with regard to positive effects of breastfeeding in other domains, this information is critical.

Action: We have now added “*Mothers were asked “Did you breastfeed your baby?” If they said “no” this was recorded as “never breastfed.” If they said “yes” they were asked “At what age did you stop breastfeeding?” with the answer recorded in months.*” (Page 6, Line 10-12)

Presumably diet was assessed at 1 year. Has the EAT score been well validated against other measures of dietary intake or quality or widely used in dietary assessments of 1 year olds? Are higher scores indicative of better or worse diet quality?

It was not clear what was meant by the following statement:

“Because of varying questionnaire responses, only the food quality was included in a quasi-quantitative score”. Please clarify.

Response: We appreciate the opportunity to clarify. The strategy of the EAT-score – i.e. categorising food items into 7 overall food groups – was based on the Youth Healthy Eating Index (10.1016/j.jada.2004.06.020), which was validated against the well-known Healthy Eating Index (HEI), developed in the US to assess adherence to dietary guidelines (i.e. “healthy diet”). Food categorisation was done under the supervision of 3 trained nutritionists for accuracy.

Action: We have now modified the text as follows: “*Diet was assessed using 24-hour maternal dietary recall [37] and categorised with nutritionist supervision according to the categories of the Youth Healthy Eating Index that reflects dietary guideline adherence in*

children [38]. Parents were asked: "Please describe what food and drink your child has eaten in the past 24 hours (please specify type of food/drink and quantity)" and space was provided for breakfast, morning snack, lunch, afternoon snack, dinner and evening snack. Because of varying questionnaire responses related to portion size, only the food quality at each meal was included in a quasi-quantitative score, the Eating Assessment in Toddlers (EAT) score [37]. In brief, seven food sub-categories - wholegrain, vegetables, fruits, meat ratio ($\frac{\text{white meat} + \text{egg} + \text{other protein sources}}{\text{red meat} + \text{processed meat}}$), dairy, snack foods, and sweetened beverages - were scored 0-10 points depending on how many times they were offered at meals during the day. Our primary outcome, the total EAT score (range 0-70), summed these numbers, treating the first five components as positive and the latter two as negative (higher score proxying healthier diet). " (Page 6, Line 14-24)

Statistics: $p < .025$ as a Bonferroni correction is only accurate if there are only two outcomes (analyses) rather than only two primary exposures.

Response: The reviewer is of course correct and we are grateful for the chance to clarify. As stated this was selected "pragmatically" due to interaction analyses as the primary outcome (requiring more power) and the likely sample size requirements. As such the p-value threshold was selected for hypothesis generating purposes which was not clear from the text. The text will be amended accordingly.

Action: We have now modified the method to specify: "Given the two primary exposures of interest (breastfeeding and year 1 diet) and the five genetic profiles with potential interaction, we pragmatically chose a significance threshold of 0.025 to limit chance findings while preserving power within our limited sample size." (Page 8, Line 14-16)

Action: We have modified the discussion to state "Our results should also be interpreted with caution as our p-value threshold for significance was pragmatically chosen to account for a limited sample size, rather than a strict Bonferroni correction." (Page 14, Line 17-18)

Why were breastfeeding data split into groups rather than retaining power by keeping the duration as a continuous variable?

Response: Although the primary analysis was done with a continuous variable for the very reason stated by the reviewer, the post hoc split was done to "to explore potential breastfeeding-sensitive age windows". An additional benefit came from the nature of the data. The breastfeeding duration variable is very right skewed, meaning those with late cessation of BF could potentially bias the results due to inflated leverage in the OLS. Using categories confirmed our results without this risk.

Action: We have now added "...and potential non-linear effects" (Page 9, Line 4)

Why were effects of breastfeeding and diet quality tested separately? It is plausible there is an additive effect of these.

Response: There very well could be and we agree that it is an interesting question. Could we for instance save the adverse phenotype produced by high CMSP-PGS x short BF through improved diet? We discussed the inclusion of such an analysis, but opted not to do

it. The reasons that we did not investigate this were four-fold. 1) The primary aim of our paper was to examine whether genetic risk interacted with early nutrition to influence lasting behaviour based on the theory of “gene-environment-developmental stage” interaction (GxExD). 2) three-way interactions are hard to communicate and understand and this is also the case for the PGSxBFXDiet 3) we worry about cutting up the sample size even more and producing an ever-increasing number of tests with decreasing n. 4) Another statistical concern arose as a minority of the children were still breastfed at the time of the EAT1-scoring. Although minimally influential on the overall score, this means, that some of the variance from the EAT1-variable would be explained by some of the variance in the BF duration introducing slight collinearity (See fig 1b) which could bias the results of the three-way interaction (breastfeeding interacting with itself).

How was the repeated nature of the outcome measure treated within the analysis?

Response: The repeated measures were accounted for in a linear mixed effects model with the IDs of participants used as the clustering variable. This allows us to include all available datapoints for each participant while being able to account for the age of assessment.

What is figure 1 b intended to show?.

Action: The legend has now been modified to state “*Correlation table of the variables used in regression.*” (See figure 1b)

It is not clear at all what the duration or longevity of the effects you have documented are because it is not clear how the behaviour outcome is treated in the analysis. If the effects are observed at 17 this is very different from observations at 2 years. Does the dietary influence occur early, triggering positive developmental cascades? Or does it manifest later, implying other mechanistic effects? Whether there were important effects that determined the difference between clinically significant problems or not was also not clearly presented. The results should present all analyses and whilst some may be presented in supplementary materials, the reader should be able to access all of the results. Currently the manuscript feels selective in its presentation, especially given the number of exploratory and post hoc analyses.

Response: We are sorry to learn that this was unclear. We have certainly tried presenting all of our analyses in a comprehensible manner and we also agree that the manuscript is dense given the many post-hoc analyses. Regarding the duration of nutritional effects, we refer the reviewer to sensitivity analyses in supplementary table 4. Here we investigated if the age of the CBCL assessment influenced the effect estimate of the nutritional measure. Overall there was no certain interaction with age, but there was a trend that the effects of diet *increased* with age. This was commented in the text of the results twice, namely “*The primary results were present in the term-born cohort and did not diminish with increasing age (sT4).*” and “*In the supplementary analysis (sT4), there was a trend towards increasing diet effect size with advanced age, but this was not confirmed in the continuous model's three-way interaction and the signal was already present at two years of age*”

Action: to highlight this we have added to the discussion “*The behaviour association was detectable at age two, was consistent across ages with no sign of effect size reduction with increasing age..*” (Page 12, Line 14-15)

Rebuttal letter

The reviewer remarks are kept in black font, and our answers and actions are provided in red, with quotes in "*black italics*".

Reviewer #2 (Remarks to the Author):

The authors have addressed most of my concerns. However, I still feel the answer to the following is unsatisfactory:

Original Query (Abstract): How confident are the authors of this conclusion: 'Breastfeeding and early diet can potentially reduce behaviour problems in later life'. It was an observational study. Perhaps more difficult children are harder to get to eat a healthy diet or maintain longer breastfeeding?

Action: We have now changed the wording to "Targeted breastfeeding and early diet interventions have the potential reduce later behaviour problems.(Page 2, Line 23-24).

This conclusion is still the same. I would recommend a re-wording or addition to the sentence that highlights the need for a trial to test this potential - indeed the conclusions within the discussion itself are more appropriately tempered.

Action: This sentence has been deleted and replaced with "*We also highlight the possibility of including genetics in early nutrition intervention trials for causal inference.*"

Typo in abstract: 'a strong signal suggested that 1 EAT1 point increase result in a reduced..' result should read resulted.

Action: As per the editor instructions, this segment needs to be in present tense, and so it now reads "*results*"